# Random Projection Against Gradient Leakage: Privacy-Preserving in Federated Learning

## Abstract

Federated learning has emerged as an effective paradigm for collaborative model training under strict data-privacy constraints. However, conventional federated learning remains vulnerable to gradient-based reconstruction attacks, which can expose sensitive information. Existing privacy-preserving methods often incur significant performance degradation. To address this challenge, we propose a algorithm that balances privacy and utility by combining Random Projection Filters (RPFs) with controlled data perturbations. Specifically, during local training, each client first injects optimized noise into the data that minimally affects model performance to enhance privacy. Simultaneously, a subset of convolutional kernels is replaced with random projection filters, which structurally randomize the network and reduce the risk of sensitive data being revealed through gradients. This combination effectively strengthens privacy while inducing only a marginal drop in model accuracy. Extensive experiments demonstrate that our approach substantially improves privacy protection while maintaining high model performance, providing a practical solution for mitigating privacy risks in federated learning.

## 1 Introduction

Federated learning McMahan et al. (2017); Kairouz et al. (2019) has gained significant attention as a distributed machine learning paradigm that enables multiple clients to collaboratively train a global model without sharing their local data. By keeping the data decentralized and only exchanging model updates, federated learning aims to preserve the privacy of individual clients while leveraging their collective knowledge. This paradigm has found applications in various domains, such as mobile computing Hard et al. (2019), healthcare Xu et al., and finance Long et al. (2020), where data privacy is of utmost importance.

However, researches have revealed significant privacy vulnerabilities in standard federated learning implementations. Gradient-based attacks Zhu et al. (2019); Geiping et al. (2020) have demonstrated the potential for adversaries to reconstruct sensitive information from the gradients exchanged during the learning process. These attacks exploit the fact that gradients contain information about the underlying training data, and by carefully analyzing the gradients, an adversary can potentially infer or reconstruct private data. Such privacy breaches pose a severe threat to the confidentiality of clients' data and undermine the trust in federated learning systems.

To address these privacy concerns, various techniques have been proposed, such as differential privacy Dwork et al. (2014); Abadi et al. (2016), secure aggregation Bonawitz et al. (2017), and homomorphic encryption Aono et al. (2017). However, these techniques often come at the cost of reduced model performance or increased computational overhead. Finding an optimal balance between privacy and utility remains a challenging problem in federated learning.

In this work, we propose **fedRPF**, a novel privacy–utility trade-off algorithm for federated learning based on data distortion techniques. fedRPF enhances privacy protection with minimal performance degradation by replacing a subset of the convolutional kernels in each client's model with random projection filters and applying controlled perturbations to the local data.

Our main contributions are as follows:

- We introduce fedRPF, an innovative algorithm that effectively balances privacy preservation and model performance in federated learning through embedded random projection filters and data distortion.

- We conduct extensive experiments on multiple benchmark datasets, demonstrating that fedRPF substantially improves privacy guarantees while incurring only a negligible cost in accuracy, outperforming traditional privacy-preserving methods.

## 2 RELATED WORK

### 2.1 PRIVACY ATTACKS ON FEDERATED LEARNING

Federated learning enables multiple data owners to collaboratively train a global model without sharing their private raw data, yet it remains vulnerable to three principal privacy attacks. First, model inversion (or data reconstruction) attacks recover training samples and their labels by analyzing client-uploaded gradients or model updates; the Deep Leakage from Gradients (DLG) algorithm of Zhu et al.Zhu et al. (2019) and its analytical improvement iDLG by Zhao et al.Zhao et al. (2020) are representative. Second, attribute inference attacks seek to infer sensitive but non-target attributes from gradients or aggregated updates—for example, the cosine-similarity matching attack of Lyu et al.Lyu & Chen (2021) and the Practical Attribute Reconstruction Attack (Cos-matching ARA) of Chen et al.Chen et al. (2024), both of which recover private attributes using only periodically shared gradients. Third, membership inference attacks determine whether a specific sample participated in training; Shokri et al.Shokri et al. (2017) first trained adversarial classifiers on model output differences, and Nasr et al.Nasr et al. (2019) quantified this threat under asynchronous and synchronous aggregation protocols, proposing a defense that combines differential privacy with regularization.

### 2.2 PRIVACY-PRESERVING TECHNIQUES AND UTILITY TRADE-OFFS

To mitigate these attacks, a variety of privacy-preserving methods have been developed, balancing protection and model utility. The most widespread approach injects differential privacy noise during aggregation: Geyer et al.Geyer et al. (2017) add Gaussian noise to achieve client-level privacy in FedAvg, while the FedNFL frameworkZhang et al. (2023) adaptively learns noise distributions to improve accuracy under a fixed privacy budget. Data-distortion techniques perturb inputs or intermediate representations, such as the Soteria mechanism of Sun et al.Sun et al. (2021), which applies controlled noise to fully connected layer activations to thwart gradient inversion, particularly under non-IID data distributions. Cryptographic solutions include Bonawitz et al.'s Practical Secure Aggregation protocolBonawitz et al. (2016), which uses secret sharing and homomorphic encryption so that the server only learns the sum of client updates, and the SMPAI framework of Phong et al.Mugunthan et al. (2019), which combines additive homomorphic encryption with asynchronous SGD to preserve accuracy with manageable communication overhead. In contrast, our work embeds random projection filters directly into the model architecture and integrates learnable noise injections, achieving a more efficient privacy–utility trade-off at its source.

## 3 BACKGROUND AND MOTIVATION

We consider a standard horizontal federated learning setting (see Appendix A.2 for formal definitions), and the notations used in this paper are defined in the AppendixA.1.

### 3.1 THREAT MODEL

Throughout this work, we adopt the widely used *semi-honest server* assumption. Specifically, the central server is assumed to faithfully execute the prescribed learning protocol, yet it is considered curious in the sense that it records and analyzes all accessible information with the intention of inferring sensitive attributes of client data. In particular, the server is assumed to observe aggregated gradients or model updates, auxiliary metadata such as timing or synchronization patterns, and possess prior knowledge of the underlying learning system, including model architecture, loss function, and training hyperparameters. These observational capabilities enable inference attacks that recon-

---

**Algorithm 1** SampleRandomProjectionFilters

---

**Input**: Number of convolutional layers $L$, number of filters per layer $N^{(l)}$, number of random projection filters per layer $N_r^{(l)}$

**Output**: Filter weights $\{F_1^{(l)}, \ldots, F_{N^{(l)}}^{(l)}\}_{l=1}^L$

1: **for** $l = 1$ **to** $L$ **do**
2:    **for** $i = 1$ **to** $N_r^{(l)}$ **do**
3:        $k \leftarrow$ kernel size of convolution filter
4:        $F_i^{(l)} \sim \mathcal{N}(0, 1/k^2)$ {Random projection filter in layer $l$}
5:    **end for**
6:    **for** $i = N_r^{(l)} + 1$ **to** $N^{(l)}$ **do**
7:        Initialize $F_i^{(l)}$ as trainable {Standard trainable convolutional filter}
8:    **end for**
9: **end for**
10: **return** $\{F_1^{(l)}, \ldots, F_{N^{(l)}}^{(l)}\}_{l=1}^L$

---

**Algorithm 2** FedRPF

---

**Input**: Global model parameters $w_t$, local dataset $\mathcal{D}_k$, number of RPFs per layer $N_r$

**Hyperparameters**: local epochs $E$, perturbation iterations $T$, model learning rate $\eta$, perturbation step $\eta_p$, noise variance $\sigma$, perturbation bound $c$

**Output**: Updated local model parameters $w_k$

1: $w_k \leftarrow w_t$ {Initialize local model with global parameters}
2: **for** $e = 1$ **to** $E$ **do**
3:    **for** each minibatch $(X, y) \in \mathcal{D}_k$ **do**
4:        $\delta^0 \sim \mathcal{N}(0, \sigma^2)$ {Initialize input perturbation}
5:        $[c_\ell, c_u] \leftarrow [c, 2c]$ {Perturbation bounds}
6:        **for** $j = 1$ **to** $T - 1$ **do**
7:            $\{F^{(l)}\}_{l=1}^L \leftarrow$ SAMPLERANDOMPROJECTIONFILTERS$(L, \{N^{(l)}\}, \{N_r^{(l)}\})$
8:            $\delta^j \leftarrow \text{Proj}_{[c_\ell, c_u]}\left(\delta^{j-1} - \eta_p \nabla_X \mathcal{L}(f_{w_k}(X + \delta^{j-1}), y)\right)$ {Projected gradient descent update for perturbation}
9:        **end for**
10:        $X' \leftarrow X + \delta^T$ {Compute perturbed input}
11:        $\{F^{(l)}\}_{l=1}^L \leftarrow$ SAMPLERANDOMPROJECTIONFILTERS$(L, \{N^{(l)}\}, \{N_r^{(l)}\})$
12:        $w_k \leftarrow w_k - \eta \nabla_{w_k} \mathcal{L}(f_{w_k}(X'), y)$ {Update model parameters}
13:    **end for**
14: **end for**
15: **return** $w_k$

---

struct or approximate private data from model updates. Accordingly, our privacy-preserving design is developed to mitigate inference risks in the presence of such a semi-honest adversary.

## 4 ALGORITHM FOR BALANCING PRIVACY AND UTILITY

In this section, we propose a novel approach that leverages random projection filters (RPFs) and optimized noise injection to balance privacy and performance in federated learning. Our framework consists of two primary components: (1) replacing a designated fraction of convolutional kernels in both server and client models with random projection filters, and (2) injecting optimized noise into the input data.

### 4.1 OVERALL FRAMEWORK

In traditional federated learning, model updates are vulnerable to gradient inversion attacks, which may lead to leakage of users' private data. Existing defenses, such as differential privacy and homomorphic encryption, either significantly degrade model performance or incur substantial com-

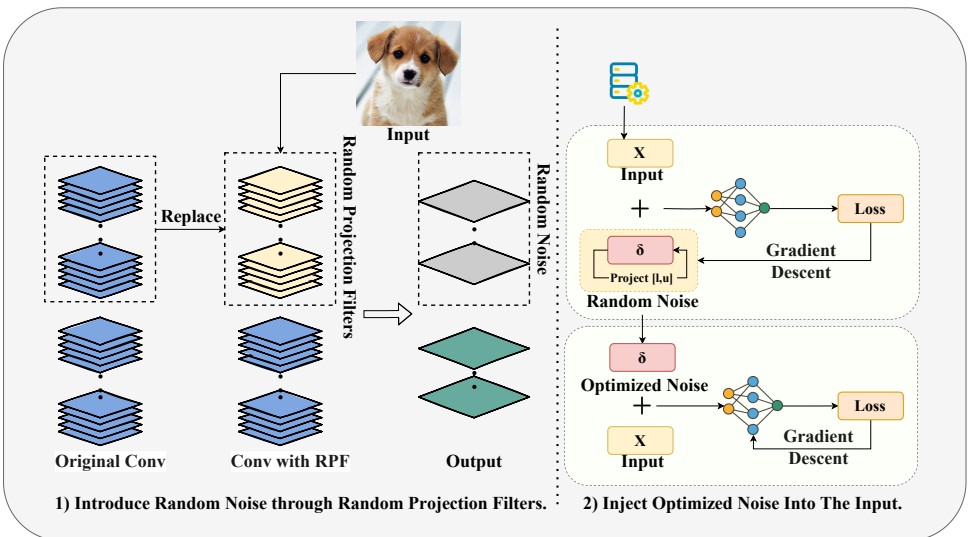

Figure 1: Overview of the FedRPF algorithm, including the process of introducing randomness via random projection filters by sampling their weights from a standard Gaussian distribution, and the process of injecting optimized noise. The weights of the random projection filters are sampled before each forward pass.

munication and computation overhead. To address these limitations, we propose FedRPF, a novel framework that combines Random Projection Filters (RPFs) with optimized noise injection, aiming to strike a balance between model utility and privacy preservation.

The overall workflow is illustrated in Figure 1. We keep the server side consistent with standard federated learning, and the pseudocode is provided in the appendix A.3. On the client side as shown in algorithm 2, two complementary privacy-enhancing mechanisms are applied: (1) a portion of convolutional kernels is replaced with random projection filters to introduce structured randomness into feature representations; (2) iterative perturbations are applied to the input data to increase the difficulty of gradient inversion attacks while minimizing the impact on model performance. Together, these mechanisms enhance privacy without compromising model effectiveness.

## 4.2 RANDOM PROJECTION MECHANISM

The random projection mechanism is inspired by the Johnson–Lindenstrauss (JL) lemma, which states that high-dimensional data can be mapped to a lower-dimensional space while approximately preserving pairwise distances. Building on this principle, a subset of convolutional kernels in each layer is replaced with random projection filters(RPFs). During each forward pass, the weights of these RPFs are resampled from a standard Gaussian distribution, thereby introducing controlled stochastic perturbations. This approach introduces structured randomness while maintaining the model's expressive capability.

Let a convolutional layer contain $N$ kernels, of which the first $N_r$ are designated as RPFs. For the input feature map $X^{(l-1)}$, the weight of the $i$-th RPF $F_i$ is sampled at each forward pass as

$$F_i \sim \mathcal{N}(0, 1/r^2), \quad i = 1, \ldots, N_r, \tag{1}$$

such that the RPFs introduce fresh stochastic perturbations during every forward propagation. The output feature map of this layer can be expressed as

$$X^{(l)} = X^{(l-1)} * \left\{ F_1, \ldots, F_{N_r}, \ F_{N_r+1}, \ldots, F_N \right\}, \tag{2}$$

where $*$ denotes the convolution operation, and $F_{N_r+1}, \ldots, F_N$ are trainable kernels. This replacement introduces structured, non-invertible randomness into part of the feature channels, improving privacy protection while maintaining model expressiveness.

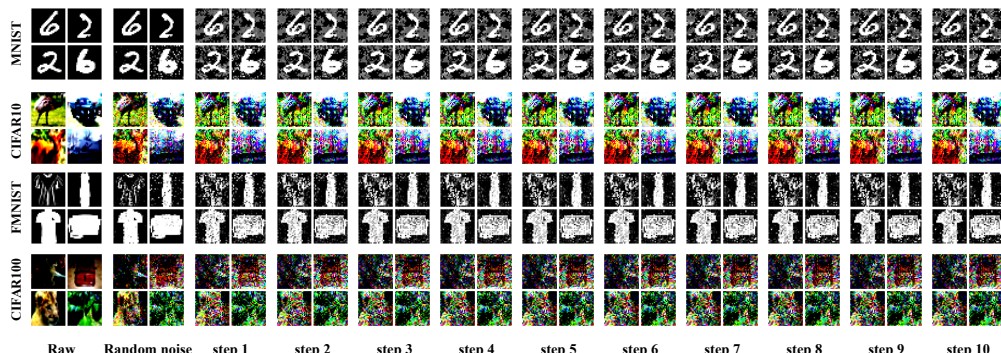

Figure 2: Illustration of optimized noise on four different datasets. Each row shows (left to right): the original input image, the image with random noise, and the images perturbed with optimized noise after 1 to 10 PGD iterations. The optimized noise selectively modifies local details while maintaining global semantic features.

### 4.3 OPTIMIZED NOISE INJECTION

We formulate the defense against gradient inversion as a min–min optimization problem: each client jointly optimizes model parameters $w$ and an input perturbation $\delta$ that lies within a constrained set. Specifically, the training objective can be expressed as

$$\min_{w} \min_{\delta \in [c_\ell, c_u]} \mathcal{L}\big(f_w(X + \delta), y\big), \tag{3}$$

where the inner minimization enforces that the perturbed input $X + \delta$ does not increase the loss, while the outer minimization updates the model parameters. **Why we adopt optimized noise?** Since random noise may disturb features critical for model performance, the goal of optimized noise is to perturb local details of the input while preserving the global structure and semantic features of the image. This ensures that the injected perturbation hinders gradient-based reconstruction without significantly degrading model utility. To obtain the perturbation $\delta$, we employ a multi-step projected gradient descent (PGD) procedure. Starting from an initial perturbation, the update rule at iteration $j$ is

$$\delta^j = \mathrm{Proj}_{[c_\ell, c_u]}\Big(\delta^{j-1} - \eta_p \nabla_X \mathcal{L}(f_w(X + \delta^{j-1}), y)\Big), \tag{4}$$

and the final perturbation is added to the input:

$$X' = X + \delta^T. \tag{5}$$

The projection step ensures that perturbations remain bounded within $[c_\ell, c_u]$, making the injected noise both controllable and task-aware.

Figure 2 provides a visual comparison between our optimized noise injection and standard random noise. We can observe that while random noise indiscriminately alters both global and local features, our optimized perturbations primarily adjust local details, preserving the overall structure and semantic content of the input images. This supports the effectiveness of our approach in defending against gradient-based reconstruction attacks while maintaining model utility. For a more detailed comparison of global and local feature distortions—illustrating how random noise disrupts global features and how optimized noise preserves them while selectively perturbing local details—please refer to Appendix A.5.

This work integrates two complementary privacy-enhancing mechanisms. First, the random projection mechanism introduces global structural perturbations by dynamically injecting structured randomness into network layers, thereby non-invasively weakening the invertibility of features. Second, the optimized noise injection mechanism generates adaptive, fine-grained perturbations that selectively interfere with gradient-based reconstruction while preserving the global semantic information of the input images. Together, these mechanisms synergistically enhance privacy protection without compromising model utility.

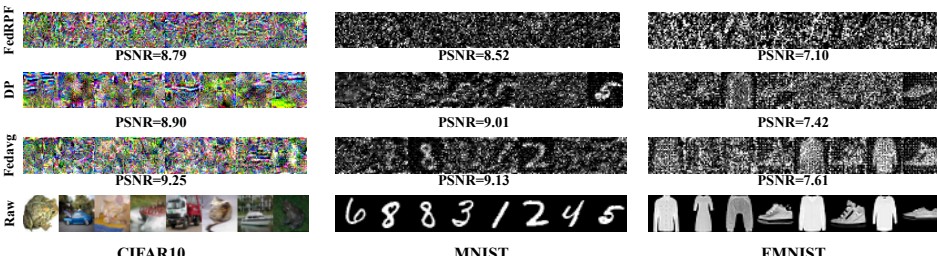

Figure 3: Privacy protection performance of DP-Laplace and fedRPF under comparable accuracy levels.

## 5 EXPERIMENT

In this section, we provide a detailed overview of our experimental setup and highlight the key results demonstrating the effectiveness of the FedRPF algorithm. Complete experimental data and additional analyses are available in the Appendix.

### 5.1 EXPERIMENT SETUP

**Datasets and Models.** We evaluate FedRPF on four datasets: MNIST, Fashion-MNIST, CIFAR-10, and CIFAR-100. For MNIST and Fashion-MNIST, we adopt the LeNet architecture; for CIFAR-10, we use ResNet-18, and for CIFAR-100, we use ResNet-34. In each convolutional layer, a proportion of the kernels determined by the RPF ratio are replaced with Random Projection Filters (RPFs). The weights of these RPFs are sampled from $\mathcal{N}(0, 1/k^2)$, where $k$ denotes the kernel size, consistent with the approach in Dong & Xu (2023). The remaining kernels remain fully trainable.

**Baselines.** We compare FedRPF against five representative federated learning methods: standard FedAvg McMahan et al. (2017), FedAvg with Laplace differential privacy (DP-Laplace) Geyer et al. (2017), FedAvg with Gaussian differential privacy (DP-Gaussian) Abadi et al. (2016), the adaptive noise-learning framework FedNFL Zhang et al. (2023), and Adaptive Local Differential Privacy (AlDP) Cui & Wu.

**Training Configuration.** For MNIST, Fashion-MNIST, and CIFAR-10, each client is allocated 2,500 training samples and trained for 30 local epochs. For CIFAR-100, each client is allocated 12,500 training samples and trained for 40 local epochs. The learning rate is set to 0.001. For the MNIST, Fashion-MNIST, and CIFAR-10 datasets, each client is assigned 2,500 training samples and performs one local update per round. For detailed experimental settings, please refer to the Appendix A.4.

**Evaluation Metrics.** Model utility is measured by classification accuracy on test sets. Privacy protection is evaluated by mounting Deep Leakage from Gradients (DLG) attacks Zhu et al. (2019) on the first client's gradient, running $T = 1600$ iterations to reconstruct images. We assess reconstruction quality using Peak Signal-to-Noise Ratio (PSNR), Mean Squared Error (MSE), and Structural Similarity Index (SSIM), where lower PSNR/SSIM and

Table 1: Performance and privacy metrics of FedAVG on MNIST, FMNIST, CIFAR10, and CIFAR100, serving as a baseline without privacy protection.

| Dataset | Acc. (%) | PSNR (↓) | SSIM (↓) | MSE (↑) |
|---|---|---|---|---|
| MNIST | 92.23 | 9.3686 | 0.0739 | 1.2711 |
| FMNIST | 80.02 | 8.7146 | 0.1092 | 1.1477 |
| CIFAR10 | 65.49 | 9.5324 | 0.0285 | 2.1147 |
| CIFAR100 | 47.96 | 9.3734 | 0.4795 | 1.9609 |

higher MSE indicate stronger privacy protection. DLG attacks are launched on rounds 11–13.

### 5.2 MAIN RESULTS

**Performance Comparison.** Figure 4 presents the PSNR and accuracy results of five privacy-preserving algorithms—FedRPF, FedNFL, ALDP, DP-Laplace, and DP-Gaussian—across four datasets under varying privacy strengths. The x-axis represents PSNR, where a lower value in-

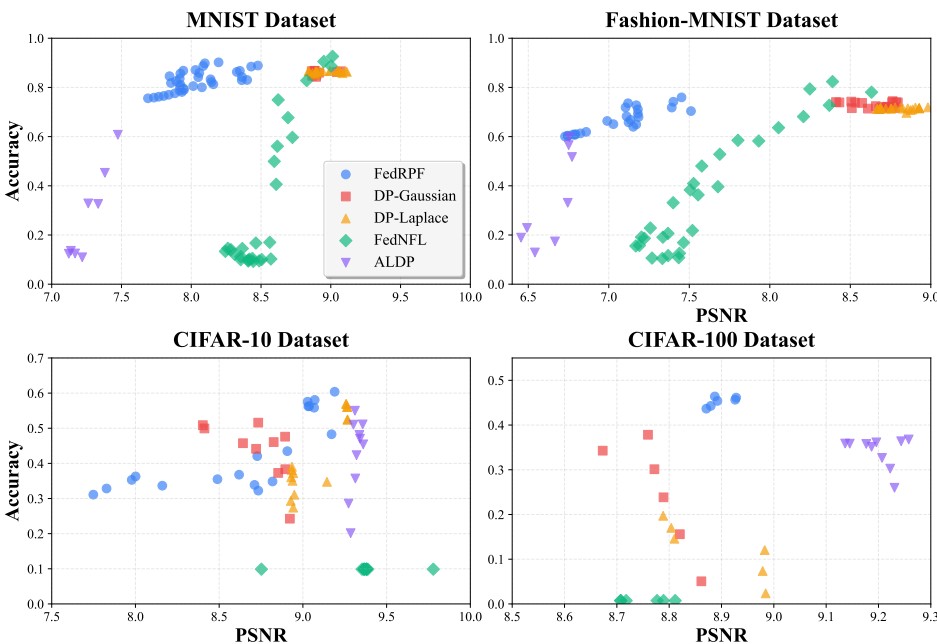

Figure 4: Privacy-Utility Trade-off Comparison between FedRPF, FedNFL, ALDP, DP-Laplace, and DP-Gaussian on four datasets.

dicates stronger privacy protection, while the y-axis represents model accuracy, with higher values indicating better performance. Therefore, algorithms positioned in the upper-left corner exhibit superior trade-offs between privacy and utility. Among the five algorithms, FedRPF, located in the upper-left region, achieves the best balance between privacy protection and model performance. Notably, FedNFL shows a clear privacy-utility trade-off on relatively simple datasets such as MNIST and FMNIST. However, on more complex datasets like CIFAR-10 and CIFAR-100, its performance deteriorates significantly. This observation suggests that the magnitude of gradient perturbation should be adjusted according to the complexity of the dataset: more complex datasets require smaller noise; otherwise, excessive noise may disrupt the original gradients and harm model training. Table 1 presents the DLG attack results under the standard FedAVG setting (without any privacy mechanism), demonstrating that all three privacy-preserving algorithms enhance federated learning privacy, albeit with varying degrees of performance degradation. The complete results can be found in the Appendix A.9.

**Privacy Protection.** We selected the DP-Laplace and FedRPF algorithms with comparable model performance to evaluate their defense against gradient reconstruction attacks, as illustrated in the figure 3. The bottom row shows the original images used for client training, while the subsequent rows display the images reconstructed from gradients by DLG attacks on FedAVG, DP-Laplace, and FedRPF, respectively. It can be observed that FedRPF provides the strongest privacy protection both visually and quantitatively, while maintaining high model performance.

**Algorithm fairness and robustness.** In figure 5, the validation accuracy of the clients (C0–C3) is plotted against training steps, as validation is performed after each minibatch, whereas the global model (Global) is plotted against communication rounds (epochs), with evaluation occurring after each aggregation. Despite the differing horizontal scales, the overall trends clearly reflect the stability and convergence of the algorithm. The client curves largely overlap, indicating that the proposed method does not introduce unfairness among clients. The global model's accuracy is consistent with, or slightly higher than, that of the clients, demonstrating effective aggregation. Even with the incorporation of random projection filters and optimized noise for privacy protection, the models maintain high accuracy and converge steadily.

Although fedRPF demonstrates a strong trade-off between privacy and utility across numerous experiments, it still has certain limitations. A detailed analysis can be found in the Appendix A.7.

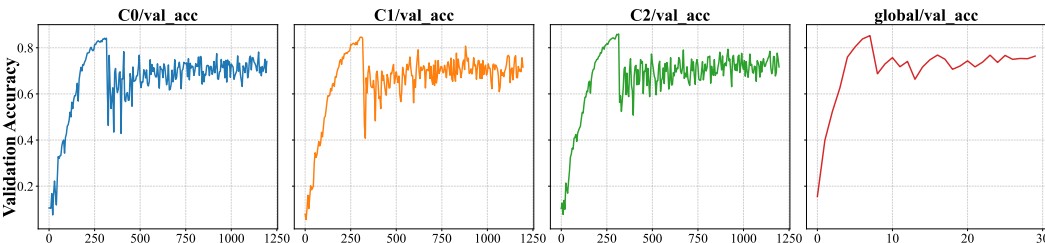

Figure 5: Validation accuracy trends for four clients (C0–C2) and the global model (Global) over training epochs. The x-axis represents the training epochs, and the y-axis shows the validation accuracy. Each subplot corresponds to a different client or the global model.

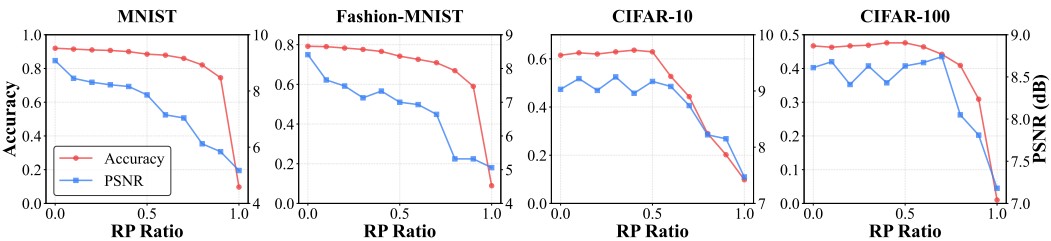

Figure 6: The trend of PSNR and accuracy (Acc) on four datasets (MNIST, Fashion-MNIST, CIFAR-10, CIFAR-100) as the RPFs ratio changes.

## 5.3 ABLATION STUDY

Since the fedRPF framework integrates both Random Projection Filters (RPFs) and optimized noise injection, we conducted a series of ablation studies to assess the necessity of each component and to examine how their parameters influence performance.

**The necessity of the Random Projection mechanism and the Optimized Noise mechanism.** We conducted four experimental settings on the MNIST dataset to evaluate the necessity of the two mechanisms. The results are shown in Table 2 and Figure 7a. Several key observations can be drawn from these results. First, comparing Experiment 1 with Experiment 4 highlights the necessity of incorporating RPF. Without RPF, although optimized noise can reduce privacy leakage to some extent, it comes at the cost of significant performance degradation. By integrating RPF, the model achieves noticeably higher accuracy, indicating that RPF effectively compensates for the negative impact of optimized noise, thereby achieving a better balance between privacy protection and model performance. Second, comparing Experiment 3 with Experiment 4 demonstrates that optimized noise provides stronger privacy protection than random noise. Specifically, while maintaining similar or even higher model accuracy, the PSNR decreases substantially, implying that the original data becomes harder for an attacker to reconstruct. This indicates that optimized noise not only works in theory but also practically enhances privacy protection in experiments.

**Effect of Random Projection Filter Ratio(RPF ratio).** We investigated the effect of increasing the random projection filter (RPF ratio) on model performance and privacy protection, using only random projection without optimization noise. As shown in the figure 6, as the RPF ratio increases, model accuracy declines, indicating performance degradation, while PSNR decreases, reflecting stronger privacy protection. When the RPF ratio is small (e.g., 0.1 0.5), the model maintains high accuracy and PSNR, suggesting sufficient feature extraction by the remaining convolutional kernels. However, as the RPF ratio continues to rise, privacy protection improves, but performance declines sharply when the RPF ratio approaches its maximum, as the remaining kernels can no longer effectively extract image features. Therefore, a high RPF ratio should be avoided to prevent significant performance loss.

**Number of Noise Optimization Iterations.** As shown in Figure 7b, increasing the number of noise optimization iterations leads to a gradual decrease in PSNR, indicating stronger privacy protection, while the model accuracy remains nearly unchanged. This trend demonstrates that optimized noise

Table 2: PSNR and accuracy under different privacy settings on MNIST.

| Setting | PSNR | MSE | SSIM | Accuracy |
|---|---|---|---|---|
| 1. No RPF & Optimized Noise | 7.96 | 1.705 | 0.030 | 0.7288 |
| 2. RPF & No Noise | 7.86 | 1.751 | 0.032 | 0.8960 |
| 3. RPF & Random Noise | 7.82 | 1.760 | 0.030 | 0.7752 |
| 4. RPF & Optimized Noise | 7.67 | 1.823 | 0.029 | 0.7712 |

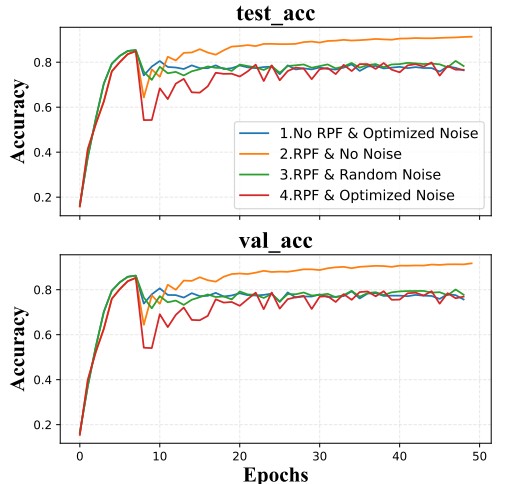

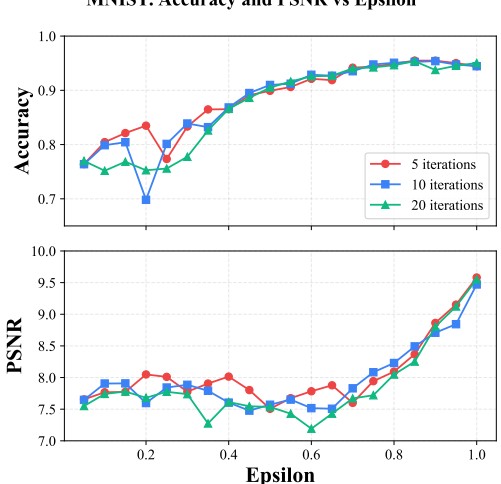

(a) Different ablation settings. The first 8 rounds serve as a warm-up phase.

(b) Different numbers of noise optimization iterations.

Figure 7: Overall results of the ablation study on fedRPF. Subfigure (a) compares test and validation accuracy under different ablation settings, while subfigure (b) illustrates the privacy–utility trade-off with varying numbers of noise optimization iterations.

can enhance privacy with minimal impact on model performance. Moreover, the effectiveness of noise optimization may also depend on factors such as the learning rate and the complexity of the dataset, which will be investigated in future work.

In summary, the ablation studies confirm that both components of the fedRPF algorithm, RPFs and optimized noise, play crucial roles. RPF is essential for mitigating the accuracy loss caused by noise injection, while optimized noise provides stronger privacy protection than random noise without significantly affecting model performance. These results highlight the complementary effects of RPF and optimized noise in achieving a robust privacy and utility trade off.

## 6 CONCLUSION

In this work, we propose a privacy-preserving algorithm, fedRPF, which aims to enhance privacy protection in federated learning without significantly compromising model performance. The algorithm achieves this by replacing a subset of convolutional kernels in the model with random projection filters and by applying optimized noise to the input data. Extensive experiments on FMNIST, MNIST, CIFAR-10, and CIFAR-100 demonstrate that our method consistently outperforms multiple baseline algorithms, achieving superior results across these datasets. However, fedRPF has certain limitations, notably higher computational and training costs on large or complex datasets due to repeated RPF sampling and noise optimization. Our future work will focus on improving efficiency while preserving strong privacy guarantees.

## 7 REPRODUCIBILITY STATEMENT

To ensure the reproducibility and verifiability of our results, we provide the full codebase, all reported results, and step-by-step usage instructions at the following anonymous link: `https://anonymous.4open.science/r/fedRPF-A74E342/README.md`. In particular, implementations of our method and all baselines are available; the hyperparameter tuning, training, and evaluation pipelines are provided; the hyperparameters used are fully documented; the datasets and their splits are included; and within a fixed software environment, the results are largely bitwise reproducible.

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

## A  APPENDIX

### A.1  NOTATION

The notations used in this paper are shown at table 3.

| Notation | Description |
|---|---|
| $K$ | Number of clients |
| $\mathcal{D}^{(k)}$ | Dataset of client $k$ |
| $|\mathcal{D}^{(k)}|$ | Dataset size of client $k$ |
| $(x_i^{(k)}, y_i^{(k)})$ | $i$-th sample in client $k$'s dataset |
| $W$ | Model parameters |
| $\mathcal{L}^{(k)}(W)$ | Local loss of client $k$ |
| $\alpha_k$ | Aggregation weight of client $k$ |
| $\delta$ | Perturbation applied to data |
| $\epsilon$ | Privacy budget / threshold |

Table 3: Key notations used in this paper.

## A.2 PROBLEM FORMULATION

We consider a horizontal federated learning (HFL) system comprising a central server and $K$ heterogeneous clients $\mathcal{K} = \{1, \ldots, K\}$. Each client $k \in \mathcal{K}$ possesses a private dataset $\mathcal{D}^{(k)} = \{(x_i^{(k)}, y_i^{(k)})\}_{i=1}^{|\mathcal{D}^{(k)}|}$, where $x_i^{(k)} \in \mathcal{X}$ denotes the input features and $y_i^{(k)} \in \mathcal{Y}$ the corresponding labels. The empirical risk minimized locally by client $k$ under model parameters $W$ is

$$\mathcal{L}^{(k)}(W) = \frac{1}{|\mathcal{D}^{(k)}|} \sum_{i=1}^{|\mathcal{D}^{(k)}|} \mathcal{L}(W, x_i^{(k)}, y_i^{(k)}), \tag{6}$$

while the federated optimization objective is defined as

$$W^* = \arg\min_W \sum_{k=1}^{K} \alpha_k \mathcal{L}^{(k)}(W), \qquad \alpha_k = \frac{|\mathcal{D}^{(k)}|}{\sum_{j=1}^{K} |\mathcal{D}^{(j)}|}, \tag{7}$$

ensuring that each client contributes proportionally to the size of its local dataset.

## A.3 ALGORITHM

---
**Algorithm 3** FedRPF - Server of FL
---
1: C: proportion of clients selected for training, K: total number of clients
2: **Server Executes:**
3: Initialize global model $w_0$             ▷ Initialize global model at round 0
4: **for** each round $t = 1, 2, \ldots$ **do**
5:    $d \leftarrow \max(C \cdot K, 1)$           ▷ Number of clients chosen each round
6:    $S_t \leftarrow$ random set of $d$ clients
7:    Compute weights $\alpha_k = \frac{|\mathcal{D}^{(k)}|}{\sum_{j \in S_t} |\mathcal{D}^{(j)}|}, \forall k \in S_t$    ▷ Data-proportional weights
8:    **for** each client $k \in S_t$ in parallel **do**
9:       $w_{t+1}^k \leftarrow \text{ClientUpdate}(k, w_t)$    ▷ Clients upload updated parameters
10:    **end for**
11:    $w_{t+1} \leftarrow \sum_{k \in S_t} \alpha_k w_{t+1}^k$    ▷ Weighted average of client parameters
12: **end for**
13: **Output:** $w_t$                   ▷ Final global model
---

The server maintains a global model and, in each round, randomly selects a subset of clients to participate in training. The server then aggregates the uploaded local updates to form the next global model. The pseudocode is given in 3.

## A.4 EXPERIMENTAL SETTING

We use a total of four clients, with the first eight training rounds serving as a warm-up phase during which no privacy protection mechanisms are applied. During rounds 9, 10, and 11, when the privacy-preserving algorithms are active, the server performs DLG attacks on the received gradients to attempt reconstruction of user data. The learning rate is set to 0.001. For the MNIST, Fashion-MNIST, and CIFAR-10 datasets, each client is assigned 2,500 training samples and performs one local update per round. For the CIFAR-100 dataset, each client is assigned 12,500 training samples and also performs one local update per round. We run our experiments on an NVIDIA A6000 GPU with 48GB of memory.

## A.5 THE IMPACT OF NOISE ON DATA'S LOCAL AND GLOBAL FEATURES

As illustrated in the figure 8, the effects of different noise injection strategies on image features are visually evident. Random noise not only perturbs local details but also disrupts the overall structure and semantic information, causing a shift in the global features of the image. In contrast, optimized noise selectively perturbs local details while largely preserving the global semantic and structural information, effectively hindering gradient-based image reconstruction without compromising the

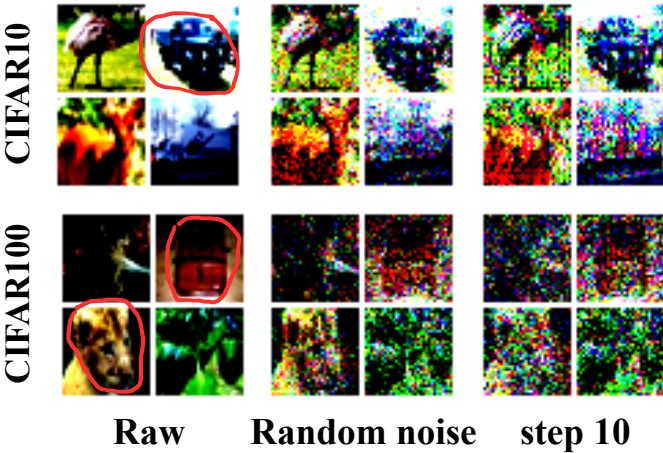

Figure 8: Comparison of perturbations on an input image. From left to right: the original image, the image with random noise, and the image with optimized noise after 10 PGD iterations. Optimized noise preserves global semantic features while selectively perturbing local details.

Table 4: Average running time of each algorithm on different datasets (unit: seconds).

| Dataset | FedRPF | FedNFL | DP-Laplace | DP-Gaussian | FedAvg | Aldp |
|---|---|---|---|---|---|---|
| MNIST | 21.60 | 10.45 | 11.26 | 15.74 | 11.47 | 11.68 |
| FMNIST | 25.83 | 10.40 | 8.37 | 8.46 | 11.65 | 10.51 |
| CIFAR-10 | 152.16 | 85.33 | 31.24 | 25.21 | 16.88 | 91.40 |
| CIFAR-100 | 1384.00 | 90.23 | 198.48 | 151.04 | 95.48 | 1111.14 |

overall content. This effect is particularly noticeable in the objects highlighted by the red boxes. These results demonstrate that optimized noise achieves a favorable balance between privacy protection and model utility.

### A.6 Ablation Study

**Computational cost.** From Table 4, it is evident that the average running time of different algorithms varies significantly across datasets. For relatively simple datasets such as MNIST and FMNIST, fedRPF incurs slightly higher training time compared with other methods (e.g., FedAvg, DP-Laplace), but the difference is modest. In contrast, for more complex datasets like CIFAR-10 and CIFAR-100, fedRPF's running time increases substantially, reaching 1384 seconds on CIFAR-100—an order of magnitude higher than FedAvg (95.48 seconds) and FedNFL (90.23 seconds). This is primarily because Random Projection Filters (RPFs) must resample weights at every forward pass, and the optimized noise mechanism requires multiple iterations per minibatch, both of which add considerable computational overhead during training. Therefore, although fedRPF achieves a favorable trade-off between privacy protection and model performance, its high computational and training cost remains a notable limitation, particularly when scaling to large or complex datasets.

**Effect of Random Projection Filter Ratio(RPF ratio).** We investigated the impact of increasing the proportion of random projection filters (RPF ratio) among the total convolutional kernels on model performance and privacy protection, using only the random projection mechanism without optimization noise. As shown in the Table 5, with the increase in RPF ratio, the model accuracy (Acc) exhibits an overall declining trend, indicating a gradual decrease in model performance. Si-

Table 5: Ablation study of RPF ratio parameter in fedRPF algorithm

| RPF ratio | 0.00 | 0.10 | 0.20 | 0.30 | 0.40 | 0.50 | 0.60 | 0.70 | 0.80 | 0.90 | 1.00 |
|---|---|---|---|---|---|---|---|---|---|---|---|
| **MNIST** | | | | | | | | | | | |
| PSNR | 9.08 | 8.45 | 8.31 | 8.22 | 8.16 | 7.86 | 7.15 | 7.04 | 6.12 | 5.84 | 5.17 |
| MSE | 1.34 | 1.53 | 1.58 | 1.61 | 1.63 | 1.75 | 2.12 | 2.12 | 2.71 | 2.92 | 3.55 |
| SSIM | 0.052 | 0.036 | 0.034 | 0.029 | 0.031 | 0.032 | 0.028 | 0.029 | 0.024 | 0.026 | 0.021 |
| Acc | 0.920 | 0.915 | 0.910 | 0.907 | 0.900 | 0.885 | 0.879 | 0.860 | 0.821 | 0.745 | 0.097 |
| **Fashion-MNIST** | | | | | | | | | | | |
| PSNR | 8.41 | 7.66 | 7.48 | 7.13 | 7.33 | 7.00 | 6.93 | 6.64 | 5.32 | 5.32 | 5.06 |
| MSE | 1.21 | 1.41 | 1.47 | 1.60 | 1.52 | 1.64 | 1.66 | 1.78 | 2.44 | 2.46 | 2.64 |
| SSIM | 0.084 | 0.046 | 0.037 | 0.038 | 0.035 | 0.036 | 0.041 | 0.036 | 0.032 | 0.032 | 0.029 |
| Acc | 0.792 | 0.790 | 0.783 | 0.776 | 0.766 | 0.742 | 0.726 | 0.709 | 0.669 | 0.590 | 0.089 |
| **CIFAR-10** | | | | | | | | | | | |
| PSNR | 9.03 | 9.22 | 9.01 | 9.25 | 8.96 | 9.17 | 9.08 | 8.74 | 8.22 | 8.15 | 7.47 |
| MSE | 2.07 | 1.97 | 2.07 | 1.95 | 2.11 | 2.00 | 2.03 | 2.19 | 2.47 | 2.53 | 3.02 |
| SSIM | 0.023 | 0.021 | 0.024 | 0.023 | 0.025 | 0.022 | 0.020 | 0.018 | 0.020 | 0.014 | 0.013 |
| Acc | 0.615 | 0.625 | 0.620 | 0.629 | 0.636 | 0.629 | 0.527 | 0.443 | 0.290 | 0.202 | 0.098 |
| **CIFAR-100** | | | | | | | | | | | |
| PSNR | 8.61 | 8.68 | 8.41 | 8.63 | 8.43 | 8.63 | 8.67 | 8.74 | 8.05 | 7.81 | 7.18 |
| MSE | 2.01 | 1.98 | 2.10 | 1.99 | 2.12 | 2.01 | 2.00 | 1.96 | 2.29 | 2.45 | 2.89 |
| SSIM | 0.030 | 0.029 | 0.029 | 0.028 | 0.029 | 0.031 | 0.032 | 0.024 | 0.024 | 0.020 | 0.015 |
| Acc | 0.467 | 0.463 | 0.467 | 0.469 | 0.476 | 0.476 | 0.464 | 0.442 | 0.409 | 0.309 | 0.010 |

multaneously, the PSNR value also decreases, signaling an enhancement in privacy protection, and highlighting the trade-off between privacy and performance.

When the RPF ratio is relatively small (e.g., 0.00 or 0.10), the model achieves higher accuracy and better PSNR, suggesting that the remaining convolutional kernels are sufficient to capture the image features. However, as the RPF ratio increases, privacy protection improves, but model performance begins to degrade. This decline becomes more pronounced when the RPF ratio approaches its maximum value, as the remaining convolutional kernels are no longer capable of effectively extracting the image features, leading to a sharp drop in performance. Therefore, the RPF ratio should not be excessively high, as too many random projection filters can cause the model to lose its ability to represent image features effectively, resulting in significant performance loss.

| rp_$\sigma^2$ | PSNR ($\downarrow$) | MSE ($\uparrow$) | SSIM ($\downarrow$) | Accuracy ($\uparrow$) |
|---|---|---|---|---|
| 0.1 | 9.85 | 1.12 | 0.179 | 0.8764 |
| 0.2 | 9.47 | 1.26 | 0.160 | 0.8620 |
| 0.3 | 9.03 | 1.40 | 0.140 | 0.8475 |
| 0.4 | 8.66 | 1.52 | 0.118 | 0.8303 |
| 0.5 | 8.39 | 1.59 | 0.096 | 0.8127 |
| 0.6 | 8.12 | 1.67 | 0.077 | 0.7904 |
| 0.7 | 7.87 | 1.75 | 0.060 | 0.7648 |
| 0.8 | 7.65 | 1.84 | 0.047 | 0.7355 |
| 0.9 | 7.44 | 1.91 | 0.036 | 0.7002 |
| 1.0 | 7.23 | 1.98 | 0.027 | 0.6624 |

Table 6: Performance on FMNIST with varying random projection strength (rp_$\sigma^2$).

**The impact of the weight sampling variance of RPFs.** Table 6 and 7 presents the results of our ablation study on the weight sampling variance of the random projection filter, where the perturbation magnitude is controlled by the random projection variance $\sigma^2$. As $\sigma^2$ increases, the privacy protection effect improves, which is specifically reflected in the decrease in both the peak signal-to-noise ratio (PSNR) and structural similarity index (SSIM) between images reconstructed via Deep Leakage from Gradients (DLG) attacks and the original images, as well as the increase in mean squared error (MSE). Importantly, the model accuracy only decreases moderately. This result demonstrates that strong privacy guarantees for the model can be achieved with minimal utility loss solely through input perturbation. Consistent with the work of Dong & Xu (2023), the sampling variance is set to the kernel size.

| rp_$\sigma^2$ | PSNR ($\downarrow$) | MSE ($\uparrow$) | SSIM ($\downarrow$) | Accuracy ($\uparrow$) |
|---|---|---|---|---|
| 0.1 | 10.75 | 0.93 | 0.198 | 0.9752 |
| 0.2 | 10.21 | 1.08 | 0.181 | 0.9718 |
| 0.3 | 9.74 | 1.24 | 0.156 | 0.9680 |
| 0.4 | 9.32 | 1.37 | 0.132 | 0.9625 |
| 0.5 | 8.96 | 1.46 | 0.109 | 0.9573 |
| 0.6 | 8.61 | 1.58 | 0.088 | 0.9496 |
| 0.7 | 8.31 | 1.67 | 0.069 | 0.9403 |
| 0.8 | 8.04 | 1.78 | 0.054 | 0.9260 |
| 0.9 | 7.79 | 1.88 | 0.041 | 0.9102 |
| 1.0 | 7.56 | 1.95 | 0.030 | 0.8920 |

Table 7: Performance on MNIST with varying random projection strength (rp_$\sigma^2$).

## A.7 LIMITATIONS

Despite achieving a favorable trade-off between privacy protection and model performance across multiple datasets, our approach has several limitations. First, for complex datasets (e.g., CIFAR-100) and deeper models, excessive perturbation or an overly high proportion of RPFs can slow down convergence or even cause training instability, indicating that noise intensity and RPF parameters must be carefully tuned according to data complexity. Second, although our method effectively defends against gradient reconstruction attacks, its robustness against other potential privacy threats, such as model inversion or membership inference attacks, requires further investigation. Finally, the computational and communication overhead cannot be ignored: as shown in Table 4, algorithms like fedRPF and ALDP incur significantly higher running times on large-scale datasets, which may limit their scalability in practical federated learning scenarios. This increased overhead primarily arises from the computational cost of noise optimization and random projection operations on complex models and large datasets.

## A.8 THE USE OF LARGE LANGUAGE MODELS (LLMS)

In this work, Large Language Models (LLMs) were used solely as a general-purpose writing assistance tool to help polish the text for clarity, grammar, and readability. All content generated by the LLMs was carefully reviewed by the authors, and only the parts considered accurate and appropriate were incorporated. The research ideation, methodology, experiments, and results were entirely developed and conducted by the authors. We take full responsibility for the content of the paper, including any portions that were influenced by LLM-assisted text editing.

## A.9 COMPLETE RESULTS

| PSNR ($\downarrow$) | MSE ($\uparrow$) | SSIM ($\downarrow$) | Accuracy ($\uparrow$) |
|---|---|---|---|
| 7.6897 | 1.8161 | 0.0309 | 0.7558 |
| 7.7325 | 1.7978 | 0.0316 | 0.7586 |
| 7.7668 | 1.7847 | 0.0311 | 0.7622 |
| 7.8026 | 1.7701 | 0.0313 | 0.7653 |
| 7.8388 | 1.7550 | 0.0311 | 0.7702 |
| 7.8444 | 1.7535 | 0.0318 | 0.8457 |
| 7.8545 | 1.7514 | 0.0310 | 0.8179 |
| 7.8828 | 1.7370 | 0.0312 | 0.7761 |
| 7.8923 | 1.7348 | 0.0319 | 0.8260 |
| 7.9166 | 1.7271 | 0.0324 | 0.7879 |
| 7.9175 | 1.7254 | 0.0324 | 0.8358 |
| 7.9192 | 1.7250 | 0.0320 | 0.8032 |
| 7.9218 | 1.7251 | 0.0325 | 0.8562 |
| 7.9227 | 1.7239 | 0.0325 | 0.8107 |
| 7.9325 | 1.7176 | 0.0304 | 0.7815 |
| 7.9431 | 1.7157 | 0.0330 | 0.8680 |
| 7.9448 | 1.7159 | 0.0326 | 0.7956 |
| 7.9479 | 1.7123 | 0.0292 | 0.7898 |
| 8.0139 | 1.6847 | 0.0311 | 0.8055 |
| 8.0300 | 1.6774 | 0.0316 | 0.8709 |
| 8.0500 | 1.6706 | 0.0315 | 0.8417 |
| 8.0575 | 1.6675 | 0.0313 | 0.8588 |
| 8.0764 | 1.6616 | 0.0317 | 0.8004 |
| 8.0857 | 1.6560 | 0.0321 | 0.8870 |
| 8.0974 | 1.6516 | 0.0317 | 0.8980 |
| 8.1383 | 1.6364 | 0.0325 | 0.8331 |
| 8.1447 | 1.6340 | 0.0325 | 0.8220 |
| 8.1584 | 1.6291 | 0.0328 | 0.8128 |
| 8.1958 | 1.6140 | 0.0323 | 0.9020 |
| 8.3264 | 1.5726 | 0.0363 | 0.8630 |
| 8.3484 | 1.5664 | 0.0367 | 0.8676 |
| 8.3620 | 1.5680 | 0.0369 | 0.8294 |
| 8.3636 | 1.5674 | 0.0367 | 0.8425 |
| 8.4005 | 1.5533 | 0.0365 | 0.8303 |
| 8.4294 | 1.5387 | 0.0396 | 0.8847 |
| 8.4775 | 1.5266 | 0.0439 | 0.8890 |

Table 8: FedRPF comprehensive results on MNIST (sorted by PSNR in ascending order).

| PSNR ($\downarrow$) | MSE ($\uparrow$) | SSIM ($\downarrow$) | Accuracy ($\uparrow$) |
|---|---|---|---|
| 6.7282 | 1.7448 | 0.0294 | 0.6007 |
| 6.7449 | 1.7398 | 0.0291 | 0.6040 |
| 6.7463 | 1.7378 | 0.0292 | 0.6024 |
| 6.7771 | 1.7278 | 0.0288 | 0.6084 |
| 6.7890 | 1.7232 | 0.0292 | 0.6070 |
| 6.7938 | 1.7210 | 0.0286 | 0.6100 |
| 6.8238 | 1.7087 | 0.0287 | 0.6132 |
| 6.8594 | 1.6949 | 0.0281 | 0.6192 |
| 6.9901 | 1.6423 | 0.0340 | 0.6636 |
| 7.0285 | 1.6281 | 0.0347 | 0.6507 |
| 7.1041 | 1.6013 | 0.0318 | 0.7186 |
| 7.1104 | 1.6025 | 0.0325 | 0.6827 |
| 7.1196 | 1.5975 | 0.0330 | 0.7357 |
| 7.1228 | 1.5983 | 0.0333 | 0.6581 |
| 7.1360 | 1.5942 | 0.0340 | 0.6679 |
| 7.1516 | 1.5888 | 0.0326 | 0.6400 |
| 7.1705 | 1.5830 | 0.0328 | 0.6495 |
| 7.1707 | 1.5807 | 0.0357 | 0.7276 |
| 7.1806 | 1.5790 | 0.0343 | 0.6933 |
| 7.1806 | 1.5793 | 0.0355 | 0.7110 |
| 7.1846 | 1.5781 | 0.0345 | 0.6798 |
| 7.3910 | 1.5083 | 0.0379 | 0.7181 |
| 7.4010 | 1.5043 | 0.0364 | 0.7418 |
| 7.4517 | 1.4885 | 0.0369 | 0.7595 |
| 7.5105 | 1.4746 | 0.0394 | 0.7038 |

Table 9: FedRPF comprehensive results on FMNIST (sorted by PSNR in ascending order).

| PSNR ($\downarrow$) | MSE ($\uparrow$) | SSIM ($\downarrow$) | Accuracy ($\uparrow$) |
|---|---|---|---|
| 7.7484 | 2.7913 | 0.0220 | 0.3110 |
| 7.8275 | 2.7628 | 0.0203 | 0.3285 |
| 7.9778 | 2.6299 | 0.0210 | 0.3530 |
| 8.0009 | 2.6130 | 0.0211 | 0.3629 |
| 8.1604 | 2.5330 | 0.0218 | 0.3365 |
| 8.4898 | 2.3308 | 0.0220 | 0.3549 |
| 8.6190 | 2.2838 | 0.0197 | 0.3678 |
| 8.7112 | 2.2165 | 0.0226 | 0.3388 |
| 8.7273 | 2.2258 | 0.0272 | 0.4206 |
| 8.7335 | 2.2063 | 0.0200 | 0.3224 |
| 8.8188 | 2.1719 | 0.0217 | 0.3485 |
| 8.9069 | 2.1215 | 0.0216 | 0.4348 |
| 9.0282 | 2.0709 | 0.0233 | 0.5755 |
| 9.0326 | 2.0620 | 0.0229 | 0.5624 |
| 9.0376 | 2.0688 | 0.0216 | 0.5628 |
| 9.0672 | 2.0525 | 0.0240 | 0.5585 |
| 9.0706 | 2.0470 | 0.0264 | 0.5806 |
| 9.1712 | 1.9992 | 0.0179 | 0.4829 |
| 9.1897 | 1.9938 | 0.0237 | 0.6039 |

Table 10: FedRPF comprehensive results on CIFAR-10 (sorted by PSNR in ascending order).

| PSNR ($\downarrow$) | MSE ($\uparrow$) | SSIM ($\downarrow$) | Accuracy ($\uparrow$) |
|---|---|---|---|
| 7.1679 | 1.5829 | 0.0302 | 0.1554 |
| 7.1900 | 1.5745 | 0.0310 | 0.1576 |
| 7.2031 | 1.5693 | 0.0315 | 0.1913 |
| 7.2209 | 1.5615 | 0.0317 | 0.1866 |
| 7.2580 | 1.5479 | 0.0315 | 0.2283 |
| 7.2686 | 1.5346 | 0.0350 | 0.1062 |
| 7.3328 | 1.5094 | 0.0312 | 0.1046 |
| 7.3360 | 1.5249 | 0.0326 | 0.1911 |
| 7.3678 | 1.4970 | 0.0326 | 0.2064 |
| 7.3688 | 1.4969 | 0.0309 | 0.1162 |
| 7.3997 | 1.5030 | 0.0349 | 0.3312 |
| 7.4342 | 1.4734 | 0.0323 | 0.1078 |
| 7.4389 | 1.4706 | 0.0325 | 0.1254 |
| 7.4643 | 1.4642 | 0.0331 | 0.1690 |
| 7.5059 | 1.4501 | 0.0334 | 0.3834 |
| 7.5194 | 1.4509 | 0.0353 | 0.2180 |
| 7.5267 | 1.4626 | 0.0388 | 0.4092 |
| 7.5548 | 1.4345 | 0.0352 | 0.3632 |
| 7.5776 | 1.4295 | 0.0359 | 0.4804 |
| 7.6778 | 1.4007 | 0.0359 | 0.3964 |
| 7.6893 | 1.3969 | 0.0408 | 0.5282 |
| 7.8021 | 1.3770 | 0.0523 | 0.5850 |
| 7.9313 | 1.3309 | 0.0525 | 0.5822 |
| 8.0543 | 1.3043 | 0.0575 | 0.6362 |
| 8.2090 | 1.2810 | 0.0739 | 0.6814 |
| 8.2485 | 1.2968 | 0.0777 | 0.7938 |
| 8.3699 | 1.2773 | 0.0862 | 0.7274 |
| 8.3896 | 1.2896 | 0.1178 | 0.8236 |
| 8.6319 | 1.1631 | 0.1022 | 0.7806 |

Table 11: FedNFL comprehensive results on FMNIST (sorted by PSNR in ascending order).

| PSNR (↓) | MSE (↑) | SSIM (↓) | Accuracy (↑) |
|---|---|---|---|
| 8.2446 | 1.6032 | 0.0295 | 0.1334 |
| 8.2623 | 1.5968 | 0.0297 | 0.1458 |
| 8.2844 | 1.5883 | 0.0306 | 0.1402 |
| 8.3098 | 1.5778 | 0.0302 | 0.1221 |
| 8.3542 | 1.5588 | 0.0346 | 0.0998 |
| 8.3554 | 1.5562 | 0.0294 | 0.1121 |
| 8.3659 | 1.5529 | 0.0296 | 0.1450 |
| 8.4083 | 1.5432 | 0.0343 | 0.0980 |
| 8.4087 | 1.5402 | 0.0342 | 0.0942 |
| 8.4177 | 1.5368 | 0.0375 | 0.1018 |
| 8.4384 | 1.5320 | 0.0330 | 0.1048 |
| 8.4445 | 1.5313 | 0.0333 | 0.0928 |
| 8.4630 | 1.5185 | 0.0314 | 0.1672 |
| 8.4867 | 1.5170 | 0.0381 | 0.0950 |
| 8.5047 | 1.5097 | 0.0386 | 0.1002 |
| 8.5626 | 1.4926 | 0.0404 | 0.1702 |
| 8.5703 | 1.4928 | 0.0398 | 0.1024 |
| 8.5948 | 1.4919 | 0.0423 | 0.4994 |
| 8.6086 | 1.4742 | 0.0371 | 0.4060 |
| 8.6170 | 1.4847 | 0.0448 | 0.5608 |
| 8.6231 | 1.4939 | 0.0484 | 0.7494 |
| 8.6933 | 1.4537 | 0.0427 | 0.6772 |
| 8.7251 | 1.4585 | 0.0454 | 0.5968 |
| 8.8274 | 1.4671 | 0.0583 | 0.8278 |
| 8.9500 | 1.3804 | 0.0528 | 0.9054 |
| 9.0016 | 1.4176 | 0.0753 | 0.8872 |
| 9.0128 | 1.4173 | 0.0699 | 0.9262 |

Table 12: FedNFL comprehensive results on MNIST (sorted by PSNR in ascending order).

| PSNR (↓) | MSE (↑) | SSIM (↓) | Accuracy (↑) |
|---|---|---|---|
| 8.7523 | 2.3249 | 0.0848 | 0.0988 |
| 9.3533 | 1.9049 | 0.0133 | 0.0988 |
| 9.3646 | 1.9029 | 0.0146 | 0.0988 |
| 9.3653 | 1.8993 | 0.0143 | 0.0988 |
| 9.3672 | 1.9032 | 0.0141 | 0.0988 |
| 9.3786 | 1.8963 | 0.0138 | 0.0988 |
| 9.3797 | 1.8955 | 0.0133 | 0.0988 |
| 9.3798 | 1.8955 | 0.0141 | 0.0988 |
| 9.3876 | 1.8900 | 0.0132 | 0.0988 |
| 9.7788 | 1.8163 | 0.0785 | 0.0988 |

Table 13: FedNFL comprehensive results on CIFAR-10 (sorted by PSNR in ascending order).

| PSNR (↓) | MSE (↑) | SSIM (↓) | Accuracy (↑) |
|---|---|---|---|
| 6.8789 | 3.2346 | 0.1162 | 0.0074 |
| 7.1036 | 3.1628 | 0.1199 | 0.0074 |
| 7.4009 | 2.7910 | 0.1229 | 0.0074 |
| 8.7073 | 1.9869 | 0.0205 | 0.0074 |
| 8.7075 | 1.9821 | 0.0190 | 0.0074 |
| 8.7178 | 1.9754 | 0.0190 | 0.0074 |
| 8.7769 | 1.9507 | 0.0206 | 0.0074 |
| 8.7892 | 1.9500 | 0.0191 | 0.0074 |
| 8.8115 | 1.9375 | 0.0210 | 0.0074 |

Table 14: FedNFL comprehensive results on CIFAR-100 (sorted by PSNR in ascending order).

| PSNR ($\downarrow$) | MSE ($\uparrow$) | SSIM ($\downarrow$) | Accuracy ($\uparrow$) |
|---|---|---|---|
| 8.6661 | 1.1620 | 0.1055 | 0.7118 |
| 8.6680 | 1.1613 | 0.1048 | 0.7132 |
| 8.6829 | 1.1535 | 0.1046 | 0.7126 |
| 8.6883 | 1.1443 | 0.1101 | 0.7150 |
| 8.6889 | 1.1605 | 0.1120 | 0.7168 |
| 8.7202 | 1.1507 | 0.1142 | 0.7138 |
| 8.7526 | 1.1503 | 0.1131 | 0.7140 |
| 8.7540 | 1.1455 | 0.1151 | 0.7136 |
| 8.7781 | 1.1327 | 0.1119 | 0.7188 |
| 8.8126 | 1.1187 | 0.1178 | 0.7184 |
| 8.8264 | 1.1271 | 0.1144 | 0.7135 |
| 8.8485 | 1.1262 | 0.1238 | 0.6964 |
| 8.8569 | 1.1077 | 0.1164 | 0.7152 |
| 8.8741 | 1.1003 | 0.1134 | 0.7079 |
| 8.8868 | 1.1174 | 0.1256 | 0.7103 |
| 8.9042 | 1.0983 | 0.1256 | 0.7181 |
| 8.9264 | 1.1018 | 0.1127 | 0.7150 |
| 8.9281 | 1.0873 | 0.1151 | 0.7144 |
| 8.9281 | 1.0917 | 0.1257 | 0.7168 |
| 8.9820 | 1.0811 | 0.1283 | 0.7189 |

Table 15: DP-Laplace comprehensive results on FMNIST (sorted by PSNR in ascending order).

| PSNR ($\downarrow$) | MSE ($\uparrow$) | SSIM ($\downarrow$) | Accuracy ($\uparrow$) |
|---|---|---|---|
| 8.8398 | 1.4092 | 0.0485 | 0.8648 |
| 8.8443 | 1.4098 | 0.0489 | 0.8654 |
| 8.8679 | 1.4012 | 0.0543 | 0.8588 |
| 8.8878 | 1.3945 | 0.0506 | 0.8516 |
| 8.8882 | 1.3934 | 0.0502 | 0.8638 |
| 8.8957 | 1.3915 | 0.0524 | 0.8649 |
| 8.8975 | 1.3905 | 0.0505 | 0.8652 |
| 8.9014 | 1.3912 | 0.0519 | 0.8594 |
| 8.9194 | 1.3786 | 0.0510 | 0.8671 |
| 8.9410 | 1.3795 | 0.0497 | 0.8658 |
| 8.9933 | 1.3648 | 0.0519 | 0.8663 |
| 8.9971 | 1.3561 | 0.0510 | 0.8670 |
| 9.0451 | 1.3488 | 0.0527 | 0.8638 |
| 9.0509 | 1.3472 | 0.0532 | 0.8670 |
| 9.0656 | 1.3427 | 0.0533 | 0.8650 |
| 9.0675 | 1.3415 | 0.0521 | 0.8626 |
| 9.0842 | 1.3381 | 0.0535 | 0.8576 |
| 9.1060 | 1.3318 | 0.0542 | 0.8666 |
| 9.1153 | 1.3296 | 0.0543 | 0.8631 |

Table 16: DP-Laplace comprehensive results on MNIST (sorted by PSNR in ascending order).

| PSNR ($\downarrow$) | MSE ($\uparrow$) | SSIM ($\downarrow$) | Accuracy ($\uparrow$) |
|---|---|---|---|
| 8.9281 | 2.1189 | 0.0278 | 0.2931 |
| 8.9301 | 2.1125 | 0.0276 | 0.3604 |
| 8.9342 | 2.1074 | 0.0272 | 0.3805 |
| 8.9346 | 2.1077 | 0.0272 | 0.3903 |
| 8.9380 | 2.1111 | 0.0271 | 0.3503 |
| 8.9421 | 2.1113 | 0.0269 | 0.2739 |
| 8.9428 | 2.1059 | 0.0276 | 0.3720 |
| 8.9489 | 2.1060 | 0.0274 | 0.3107 |
| 9.1437 | 2.0134 | 0.0267 | 0.3474 |
| 9.2585 | 1.9490 | 0.0115 | 0.5691 |
| 9.2585 | 1.9490 | 0.0115 | 0.5691 |
| 9.2644 | 1.9466 | 0.0116 | 0.5602 |
| 9.2644 | 1.9466 | 0.0116 | 0.5602 |
| 9.2664 | 1.9457 | 0.0115 | 0.5244 |
| 9.2664 | 1.9457 | 0.0115 | 0.5244 |

Table 17: DP-Laplace comprehensive results on CIFAR-10 (sorted by PSNR in ascending order).

| PSNR ($\downarrow$) | MSE ($\uparrow$) | SSIM ($\downarrow$) | Accuracy ($\uparrow$) |
|---|---|---|---|
| 8.6730 | 1.9853 | 0.0129 | 0.3427 |
| 8.7595 | 1.9555 | 0.0128 | 0.3783 |
| 8.7718 | 1.9507 | 0.0122 | 0.3012 |
| 8.7892 | 1.9434 | 0.0117 | 0.2384 |
| 8.8204 | 1.9315 | 0.0117 | 0.1559 |
| 8.8615 | 1.9150 | 0.0119 | 0.0506 |

Table 18: DP-Laplace comprehensive results on CIFAR-100 (sorted by PSNR in ascending order).

| PSNR ($\downarrow$) | MSE ($\uparrow$) | SSIM ($\downarrow$) | Accuracy ($\uparrow$) |
|---|---|---|---|
| 8.4110 | 1.2133 | 0.0954 | 0.7404 |
| 8.4323 | 1.2081 | 0.0955 | 0.7390 |
| 8.5060 | 1.1977 | 0.0903 | 0.7424 |
| 8.5093 | 1.2033 | 0.1003 | 0.7157 |
| 8.5282 | 1.1842 | 0.1022 | 0.7384 |
| 8.5729 | 1.1794 | 0.0980 | 0.7370 |
| 8.6092 | 1.1679 | 0.1082 | 0.7138 |
| 8.6586 | 1.1674 | 0.1065 | 0.7239 |
| 8.6806 | 1.1594 | 0.1043 | 0.7122 |
| 8.6922 | 1.1477 | 0.1150 | 0.7193 |
| 8.7186 | 1.1401 | 0.1113 | 0.7158 |
| 8.7232 | 1.1418 | 0.1131 | 0.7216 |
| 8.7319 | 1.1366 | 0.1153 | 0.7209 |
| 8.7599 | 1.1291 | 0.1106 | 0.7395 |
| 8.7631 | 1.1275 | 0.1087 | 0.7442 |
| 8.7697 | 1.1295 | 0.1182 | 0.7171 |
| 8.7780 | 1.1336 | 0.1115 | 0.7183 |
| 8.7850 | 1.1224 | 0.1092 | 0.7395 |
| 8.7952 | 1.1181 | 0.1066 | 0.7392 |

Table 19: DP-Gaussian comprehensive results on FMNIST (sorted by PSNR in ascending order).

| PSNR (↓) | MSE (↑) | SSIM (↓) | Accuracy (↑) |
|---|---|---|---|
| 8.8634 | 1.3949 | 0.0504 | 0.8658 |
| 8.8668 | 1.3970 | 0.0508 | 0.8475 |
| 8.8745 | 1.3918 | 0.0506 | 0.8610 |
| 8.8770 | 1.3931 | 0.0511 | 0.8562 |
| 8.8804 | 1.3919 | 0.0505 | 0.8539 |
| 8.8822 | 1.3896 | 0.0516 | 0.8676 |
| 8.8835 | 1.3905 | 0.0503 | 0.8665 |
| 8.8947 | 1.3853 | 0.0509 | 0.8431 |
| 8.8969 | 1.3879 | 0.0515 | 0.8516 |
| 8.9169 | 1.3783 | 0.0523 | 0.8599 |
| 9.0434 | 1.3500 | 0.0557 | 0.8624 |
| 9.0699 | 1.3420 | 0.0556 | 0.8657 |

Table 20: DP-Gaussian comprehensive results on MNIST (sorted by PSNR in ascending order).

| PSNR (↓) | MSE (↑) | SSIM (↓) | Accuracy (↑) |
|---|---|---|---|
| 8.4025 | 2.4070 | 0.0355 | 0.5088 |
| 8.4120 | 2.4008 | 0.0347 | 0.4994 |
| 8.6421 | 2.2721 | 0.0301 | 0.4574 |
| 8.7206 | 2.2205 | 0.0252 | 0.4414 |
| 8.7332 | 2.2237 | 0.0298 | 0.5159 |
| 8.8256 | 2.1741 | 0.0222 | 0.4606 |
| 8.8526 | 2.1561 | 0.0248 | 0.3729 |
| 8.8938 | 2.1529 | 0.0292 | 0.4760 |
| 8.8949 | 2.1448 | 0.0248 | 0.3834 |
| 8.9221 | 2.1284 | 0.0255 | 0.2426 |

Table 21: DP-Gaussian comprehensive results on CIFAR-10 (sorted by PSNR in ascending order).

| PSNR (↓) | MSE (↑) | SSIM (↓) | Accuracy (↑) |
|---|---|---|---|
| 8.6730 | 1.9853 | 0.0129 | 0.3427 |
| 8.7595 | 1.9555 | 0.0128 | 0.3783 |
| 8.7718 | 1.9507 | 0.0122 | 0.3012 |
| 8.7892 | 1.9434 | 0.0117 | 0.2384 |
| 8.8204 | 1.9315 | 0.0117 | 0.1559 |
| 8.8615 | 1.9150 | 0.0119 | 0.0506 |

Table 22: DP-Gaussian comprehensive results on CIFAR-100 (sorted by PSNR in ascending order).

| PSNR (↓) | MSE (↑) | SSIM (↓) | Accuracy (↑) |
|---|---|---|---|
| 6.455 | 1.875 | 0.0396 | 0.1893 |
| 6.4918 | 1.8659 | 0.0399 | 0.2288 |
| 6.5412 | 1.8499 | 0.034 | 0.129 |
| 6.6658 | 1.7804 | 0.0383 | 0.1739 |
| 6.745 | 1.7523 | 0.0407 | 0.331 |
| 6.7507 | 1.7278 | 0.0395 | 0.5655 |
| 6.7548 | 1.7277 | 0.0413 | 0.5977 |
| 6.7716 | 1.7225 | 0.0384 | 0.5172 |

Table 23: ALDP Experimental Results on the FMNIST Dataset

| PSNR ($\downarrow$) | MSE ($\uparrow$) | SSIM ($\downarrow$) | Accuracy ($\uparrow$) |
|---|---|---|---|
| 7.1224 | 2.0996 | 0.0271 | 0.1239 |
| 7.1374 | 2.0995 | 0.031 | 0.1357 |
| 7.1673 | 2.0987 | 0.027 | 0.1249 |
| 7.2185 | 2.0615 | 0.027 | 0.1094 |
| 7.2617 | 2.0031 | 0.0299 | 0.3284 |
| 7.3327 | 1.9734 | 0.0299 | 0.3263 |
| 7.3812 | 1.9426 | 0.0312 | 0.4529 |
| 7.474 | 1.9032 | 0.0314 | 0.6076 |

Table 24: ALDP Algorithm Experimental Results on MNIST Dataset

| PSNR ($\downarrow$) | MSE ($\uparrow$) | SSIM ($\downarrow$) | Accuracy ($\uparrow$) |
|---|---|---|---|
| 9.2728 | 1.9689 | 0.0112 | 0.2858 |
| 9.2853 | 1.9678 | 0.0118 | 0.2014 |
| 9.3017 | 1.9666 | 0.0121 | 0.5310 |
| 9.3110 | 1.9621 | 0.0124 | 0.5499 |
| 9.3131 | 1.9549 | 0.0120 | 0.3571 |
| 9.3204 | 1.9585 | 0.0118 | 0.4234 |
| 9.3375 | 1.9558 | 0.0117 | 0.4810 |
| 9.3410 | 1.9556 | 0.0119 | 0.4703 |
| 9.3572 | 1.9517 | 0.0122 | 0.5110 |
| 9.3603 | 1.9475 | 0.0117 | 0.4542 |

Table 25: ALDP comprehensive results on CIFAR-10 (sorted by PSNR in ascending order).

| PSNR ($\downarrow$) | MSE ($\uparrow$) | SSIM ($\downarrow$) | Accuracy ($\uparrow$) |
|---|---|---|---|
| 9.1363 | 1.7742 | 0.0123 | 0.3587 |
| 9.1451 | 1.7716 | 0.0126 | 0.3584 |
| 9.1766 | 1.7581 | 0.0125 | 0.3578 |
| 9.1868 | 1.7600 | 0.0134 | 0.3514 |
| 9.1959 | 1.7536 | 0.0121 | 0.3613 |
| 9.2068 | 1.7510 | 0.0129 | 0.3265 |
| 9.2224 | 1.7471 | 0.0128 | 0.3026 |
| 9.2306 | 1.7448 | 0.0131 | 0.2598 |
| 9.2433 | 1.7351 | 0.0120 | 0.3638 |
| 9.2578 | 1.7323 | 0.0127 | 0.3678 |

Table 26: ALDP comprehensive results on CIFAR-100 (sorted by PSNR in ascending order).

