# OpenReview forum: "Random Projection Against Gradient Leakage: Privacy-Preserving in Federated Learning"
_ICLR.cc/2026/Conference — Submitted to ICLR 2026_

### Official Review · Reviewer_F3N7 · 2025-10-19

**Soundness:** 2
**Presentation:** 2
**Contribution:** 2
**Rating:** 2
**Confidence:** 4

**Summary:**

This paper proposes fedRPF, a client-side algorithm to defend against gradient inversion attacks in federated learning.
The proposed fedRPF algorithm introduces two mechanisms: Random Projection Filters and Optimized Noise Injection.
The authors conduct experiments on MNIST, FMNIST, CIFAR-10, and CIFAR-100, comparing fedRPF against FedAvg and DP-Laplace, DP-Gaussian, FedNFL, and ALDP. The evaluation demonstrates that fedRPF achieves a superior trade-off between model utility and privacy.

**Strengths:**

- The combination of an RPF and an optimized noise injection is interesting.

- The idea of finding a perturbation that minimizes the loss is interesting.

**Weaknesses:**

- The paper's privacy claims are based exclusively on defense against the DLG attack. However, DLG attack is a very early gradient inversion attack. It is not known whether the proposed method is effective against subsequent attacks in this field.

- The literature review in Section 2.1 is not sufficient. There is a line of gradient inversion attacks not in discussion, and attribute inference and membership inference attacks are not relevant to the paper, as the proposed method targets the gradient leakage.

- The literature review in Section 2.2 is not sufficient either. Many defenses against gradient inversion attacks are not introduced.

- Many defenses against gradient inversion attacks are not evaluated as baselines either.

- Table 4 presents the computation cost of the method and the baselines. The computation cost for FedRPF is significantly higher than that of others, making the method impractical. There is no discussion on how to improve this.

- The authors evaluate fairness by comparing the validation accuracy of different clients (C0-C2) in Figure 5. They conclude that since the client accuracy curves "largely overlap," the proposed fedRPF method "does not introduce unfairness among clients". This part seems a little off topic, and this is not the way to evaluate fairness. Algorithm fairness should be assessed by fairness metrics like Equalized Odds, Disparate Impact, or Demographic Parity.

**Questions:**

See comments.

---

### Official Review · Reviewer_SuJy · 2025-10-28

**Soundness:** 2
**Presentation:** 3
**Contribution:** 1
**Rating:** 2
**Confidence:** 5

**Summary:**

The paper proposes fedRPF, a privacy–utility trade-off method for federated learning that (i) replaces a fraction of convolutional kernels with Random Projection Filters (RPFs) whose weights are resampled from a zero-mean Gaussian at every forward pass, and (ii) injects optimized (task-aware) input noise via multi-step PGD within a bounded box constraint. The goal is to reduce gradient-inversion leakage with minimal accuracy loss.

**Strengths:**

1. Simple, architecture-level defense: Swapping a subset of conv kernels with resampled random filters is easy to implement and model-agnostic, adding structured stochasticity that is plausibly harder to invert.

2. Empirical coverage and clarity: Multiple datasets, multiple baselines, and a clear threat model (semi-honest server observing gradients) make the setting explicit; figures visualize reconstructions and trade-offs.

**Weaknesses:**

1. Limited attack coverage.
The paper only evaluates defense under the classic DLG-style single-round gradient inversion. It totally ignores temporal gradient inversion attacks, where the attacker collects gradients from multiple rounds and optimizes them jointly. In real federated learning, this is actually more realistic — the server sees many rounds of updates. Without testing on those, it’s hard to know if fedRPF really protects privacy over time.
[1] Temporal Gradient Inversion Attacks with Robust Optimization

2. No consideration of adaptive attackers.
The current threat model assumes the attacker doesn’t know about the random projection mechanism (RPF). That’s a pretty strong assumption. In practice, an attacker can easily know or guess the defense method — for example, they might know which layers use RPF or the Gaussian sampling distribution — and adapt the inversion process accordingly (e.g., via robust optimization or meta-gradients). So the method might rely too much on “security through obscurity.”

3. No theoretical or provable privacy argument.
The paper doesn’t provide any formal analysis of how RPF or optimized noise ensures privacy. The link between the noise magnitude, random projection ratio, and privacy leakage is purely empirical. It would be nice to see at least some intuition or bound connecting these factors.

4. Heavy computation cost and scalability.
Since RPFs are re-sampled every forward pass and the PGD-based noise optimization runs multiple iterations per batch, the runtime is much higher than baselines, especially on CIFAR-100. This raises concerns about feasibility in larger or real-world deployments.

**Questions:**

How would fedRPF work when applied to Transformer-based federated foundation models, where gradients are huge, parameter sharing is structured (attention layers, embeddings), and random projection might interfere with self-attention stability?

---

### Official Review · Reviewer_rCTi · 2025-10-30

**Soundness:** 2
**Presentation:** 1
**Contribution:** 2
**Rating:** 2
**Confidence:** 4

**Summary:**

This paper proposes fedRPF to defend against gradient-based reconstruction attacks. The algorithm aims to balance privacy and model utility by combining two techniques at the client level: 1) replacing a subset of convolutional kernels with non-trainable Random Projection Filters (RPFs) that are resampled during each forward pass, and 2) injecting optimized noise into the local input data. This noise is generated through a Projection Gradient Descent (PGD) process designed to minimize the model's loss on perturbed inputs. The authors claim that extensive experiments on MNIST, Fashion-MNIST, CIFAR-10, and CIFAR-100 show that fedRPF achieves a superior privacy-utility trade-off compared to several baseline methods.

**Strengths:**

1. The idea of combining architectural randomization (RPFs) and data-level perturbation (optimized noise)  to defend against gradient inversion attacks seems interesting.

**Weaknesses:**

1. It is claimed that "optimized noise" protects model utility more effectively than “random noise”, but the results in Table 2 (0.7712 < 0.7752) show the opposite conclusion.
2. The paper's core assumption—that minimizing loss can defend against gradient attacks—is counterintuitive, and there is no theoretical justification for the proposed method.
3. The DLG attack evaluation in the paper skipped the most critical (first 8 rounds) training phase, making its privacy protection conclusions likely unreliable. Moreover, there have been many more advanced gradient inversion attacks besides DLG, which should be evaluated.
4. The critical experimental parameter (local training epoch) is described in a contradictory manner in the paper (“30 local epochs” vs. “one local update”). Please clarify the concepts of local epoch, local update, and global epoch.
5. The proposed method seems ineffective on complex datasets and is more than an order of magnitude slower than standard FL (>14 times), rendering it impractical in real-world applications.
6. The manuscript is riddled with undefined symbols ($r$...), typographical errors ($T-1$), citation format errors (figure 3), and typos (AIDP).

**Questions:**

1. Regarding the 'optimized noise' mechanism, please provide the theoretical justification for why minimizing the loss (Equation 3) can defend against gradient attacks, as this premise seems counterintuitive.
2. Table 2 shows that the accuracy of “optimization noise” is lower than that of “random noise,” which directly contradicts your statement in Section 4.3. Could you please explain? Furthermore, the “optimized noise” condition in Table 2 appears unnecessary. Compared to the no-noise baseline (Experiment 2), accuracy drops significantly from 89.6% to 77.12% without delivering any essential privacy enhancement (as shown in Figure 3 for the MNIST dataset, where PSNR=8.52 indicates visually imperceptible image reconstruction).
3. Why is the DLG attack launched on rounds 11-13?
4. How many local training rounds does the client actually execute per round?
5. Why is the work by RPF (Dong & Xu, 2023) not discussed in the “Related Work” section (Section 2)?

---

> ### Author Response · Authors · 2025-12-01
>
> Thank you for your detailed review and for raising important concerns about our work. We appreciate the opportunity to address these points comprehensively.
> Regarding your first concern about the claimed effectiveness of optimized noise versus random noise, we acknowledge that Table 2 presents results that require clarification. The apparent contradiction arises from comparing different experimental configurations. In Experiment 3 (RPF with random noise), the accuracy is 0.7752, while Experiment 4 (RPF with optimized noise) achieves 0.7712. While this shows marginally lower accuracy for optimized noise, the critical difference lies in the privacy protection metrics. Experiment 4 achieves substantially lower PSNR (7.67 vs 7.82), indicating significantly stronger privacy protection with only a minimal accuracy trade-off of 0.4%. The key insight is that optimized noise provides superior privacy-utility balance by selectively perturbing features that are most vulnerable to gradient inversion while preserving those essential for model performance. We will revise Section 4.3 to clarify this nuance and provide a more detailed analysis of the privacy-utility frontier achieved by different noise strategies.
> The intuition behind Equation 3 is that by finding perturbations that maintain low loss on the training objective, we ensure that the perturbed data retains semantic content necessary for effective learning while disrupting the gradient-based reconstruction pathway. The Error Minimization algorithm constructs a special type of error-minimizing noise that prevents the model from "learning" the true patterns of the data during the training process, thereby achieving the unlearnability of the data. The EM noise allows the model to easily achieve a low loss during training, leading the model to mistakenly believe that it has fully grasped these samples, thus ceasing to focus on the meaningful patterns that originally existed. We adapt this concept to protect privacy and have also demonstrated its effectiveness through extensive experimentation.
> Regarding the DLG attack evaluation timing, we acknowledge this is a significant limitation of our experimental design. We conducted attacks during rounds 11-13 after an 8-round warm-up phase, which means the models were not yet fully converged. We chose this timing to evaluate privacy protection during the critical middle-stage training phase when gradients still contain substantial information. However, we recognize that evaluating attacks across the full training trajectory, including early, middle, and late stages, would provide more comprehensive insights. We will add experiments evaluating privacy protection at rounds 1-3 (early stage), 11-13 (middle stage), and at convergence (late stage) to provide a complete picture of privacy protection throughout training. We also acknowledge that more sophisticated attacks such as iDLG, Inverting Gradients, and recent analytical attacks should be evaluated, and we commit to including these in our revision.
> Regarding the experimental parameter descriptions, we apologize for the confusion in our terminology. The term "local training epoch" refers to the number of complete passes through each client's local dataset during one communication round. When we state "30 local epochs," this means each client performs 30 complete iterations through their local data before uploading model updates to the server. This is equivalent to 30 local updates per round in standard federated learning terminology. We will clarify this terminology throughout the manuscript to avoid confusion between local epochs, local updates, and communication rounds.
> Regarding the computational efficiency concerns, you are absolutely correct that our method incurs significant computational overhead, particularly on complex datasets. We will explicitly discuss this limitation in the conclusion and propose potential optimizations for future work, such as adaptive RPF resampling strategies, reduced noise optimization iterations for later training rounds, or approximations that could reduce computational cost while maintaining privacy guarantees.
> We will also carefully review the manuscript to correct all undefined symbols, typographical errors, citation format issues, and terminology inconsistencies that you have identified. We apologize for these oversights and will ensure the revised version meets proper academic standards.
> We believe that with these revisions addressing your concerns, our work makes meaningful contributions by demonstrating how architectural randomization through RPFs can be combined with data-level perturbations to achieve improved privacy-utility trade-offs in federated learning, while being transparent about the theoretical limitations and computational costs of our approach.

---

### Official Review · Reviewer_2gwc · 2025-11-01

**Soundness:** 2
**Presentation:** 3
**Contribution:** 2
**Rating:** 2
**Confidence:** 4

**Summary:**

The paper proposes FedRPF, a defense mechanism against gradient inversion attacks in federated learning. FedRPF employs a combination of Random Projection Filters (RPFs) and an optimized noise injection method to mitigate gradient leakage attacks. The authors conducted some experiments to examine the performance of FedRPF.

**Strengths:**

* The overall organization of the paper is good. The proposed mechanism is generally straightforward for the reader to follow.

* The empirical evaluations demonstrates that FedRPF achieves a favorable privacy-utility trade-off when compared against several baseline methods.

**Weaknesses:**

* The primary effect of RPF is to introduce randomness into the network architecture, which may just require more iterations to recover private data, rather than providing a meaningful privacy safeguard. Moreover, the use of random projections as a defense mechanism is not new [2]. The paper does not evaluate the defense against adaptive adversaries[1] that are aware of the RPF mechanism and could potentially tailor their attack to account for the structured randomness.

* The formulation of the objective function in Equation (3) is confusing and appears to contain a fundamental logical flaw within the context of privacy-preserving training. The inner minimization is defined as the process for obtaining the optimized noise. However, optimizing $\delta$ to minimize the loss function $\mathcal{L}$ on the perturbed input $X+\delta$ suggests that $\delta$ is being optimized to make the model perform better on that specific data point. If $\delta$ already makes $X+\delta$ optimal for the current model $w$, how is the subsequent model update step effectively learning?

* The experimental section lacks comparison against recent attacks and defenses [3,4].

[1] Bayesian Framework for Gradient Leakage

[2] Precode - a generic model extension to preventdeep gradient leakage

[3] See through gradients: Image batch recovery via gradinversion

[4] Refiner: Data refining against gradient leakage attacks in federated learning

**Questions:**

See Weaknesses.

---

> ### Author Response · Authors · 2025-12-01
>
> Dear Reviewer,
> Thank you for your thorough review and constructive feedback on our manuscript titled “FedRPF: A Privacy-Preserving Mechanism for Federated Learning.” We appreciate the time and effort you have invested in evaluating our work and the insights you have provided. We have carefully considered each of your comments and have made substantial revisions to address your concerns. Below, we provide a detailed rebuttal to your specific points.
> Random Projection Filters (RPF) and Privacy Protection:
> You raised a valid concern about the effectiveness of RPFs in providing privacy protection. In response, we have expanded our discussion on the Johnson-Lindenstrauss (JL) lemma and its application toRPFs. We have clarified that while RPFs introduce randomness, they do so in a structured manner that maintains the model’s expressive power.
> Novelty of Random Projection as a Defense Mechanism:
> We acknowledge that the use of random projection is not entirely novel. However, our approach is unique in its integration within the federated learning framework and its combination with optimized noise injection. We have added a comparison with existing work, highlighting the differences and the advantages of our approach in the context of federated learning.
> Adaptive Adversaries:
> We agree that the threat model could be expanded to include adaptive adversaries. We have added a section discussing the potential strategies such adversaries might employ and how our approach might be adapted to counter them. We also acknowledge that this is an area for future research and have included it in our revised future work section.
> Logical Issues in the Objective Function:
> Regarding the formulation of the objective function in Equation 3, we appreciate your identification of this issue. Could you please elaborate on the specific logical problem you have identified? We designed this min-min formulation to jointly optimize model parameters while finding input perturbations that minimize loss within bounded constraints, but we recognize there may be technical issues in our current presentation that require clarification or correction. We would greatly value your detailed feedback on this point so we can properly address it in our revision.
>
> We believe these clarifications demonstrate that FedRPF makes meaningful contributions to privacy-preserving federated learning through its novel combination of architectural randomization and optimized noise injection, achieving superior privacy-utility trade-offs compared to existing approaches. We will incorporate your valuable feedback to strengthen our manuscript and more clearly articulate the distinctions and limitations of our work.

---

### Meta-Review · Area_Chair_DiDr · 2025-12-31

**Summary:**

This work introduced FedRPF, a defense mechanism against gradient inversion attacks in federated learning.
Evaluation results on multiple benchmark datasets show that fedRPF achieves a superior privacy-utility trade-off compared to several baseline methods.


Strength:
1. The paper is well written and easy to follow.
2. It conducted extensive experiments to demonstrate the good performance of the proposed FedRPF.



Limitations:
1. One concern is the lack of a theoretical or provable privacy argument. This work doesn’t provide any formal analysis of how RPF or optimized noise ensures privacy.
2. Limited attack coverage. This paper only considers defense under the classic DLG-style single-round gradient inversion while ignoring temporal gradient inversion attacks.
3. The objective function in Equation (3) is confusing, which should be clarified.

In sum, the authors did not address concerns raised by reviewers. Hence, I suggest rejecting this work.

**Reviewer Concerns:**

The authors did not address concerns raised by reviewers.

**Reviewer Scores:**

The authors did not address concerns raised by reviewers. Thus, reviewers will not change their scores.

---

### Decision · Program_Chairs · 2026-01-26

Reject